

**Into the Noddyverse: A massive data store of 3D geological models for**
**Machine Learning & inversion applications**
Mark Jessell[1,5]; Jiateng Guo[2]; Yunqiang Li[2]; Mark Lindsay[1,5]; Richard Scalzo[3,5]; Jérémie Giraud[1];
Guillaume Pirot[1,5]; Ed Cripps[4,5]; Vitaliy Ogarko[1,5]
[1] Mineral Exploration Cooperative Research Centre, Centre for Exploration Targeting, The University of Western Australia,
Perth, Australia.
[2] College of Resources and Civil Engineering, Northeastern University, Shenyang, China.
[3] School of Mathematics and Statistics, University of Sydney, Sydney, Australia
[4] Department of Mathematics and Statistics, The University of Western Australia, Perth, Australia.
[5] ARC Centre for Data Analytics for Resources and Environments (DARE)
*Correspondence to*: Mark Jessell (mark.jessell@uwa.edu.au)



**Abstract**
Unlike some other well-known challenges such as facial recognition, where Machine Learning and Inversion algorithms are
widely developed, the geosciences suffer from a lack of large, labelled datasets that can be used to validate or train robust
Machine Learning and inversion schemes. Publicly available 3D geological models are far too restricted in both number and
the range of geological scenarios to serve these purposes. With reference to inverting geophysical data this problem is further
exacerbated as in most cases real geophysical observations result from unknown 3D geology, and synthetic test datasets are
often not particularly geological, nor geologically diverse. To overcome these limitations, we have used the Noddy modelling
platform to generate one million models, which represent the first publicly accessible massive training set for 3D geology and
resulting gravity and magnetic datasets. This model suite can be used to train Machine Learning systems, and to provide
comprehensive test suites for geophysical inversion. We describe the methodology for producing the model suite, and discuss
the opportunities such a model suit affords, as well as its limitations, and how we can grow and access this resource.

## 1 Introduction

Although it has become the focus of intense research activity in recent times, with more papers published in the five years prior to 2018 than all years before that combined, Machine Learning (ML) techniques applied to geoscience problems dates back to the middle of the last century (see Van der Baan and Jutten, 2000, and Dramsch, 2020, for reviews). ML methods are being applied to a whole range of geological and geophysical problems, but many of these studies face common challenges due to the nature of geoscientific datasets. Karpatne et al. (2017) summarise the principal challenges as follows:

i. Objects with Amorphous Boundaries- the form, structure and patterns of geoscience objects are much more complex than those found in discrete spaces that ML algorithms typically deal with, consisting of both changes in topology and dimensionality of geoscience objects with time.

ii. Spatio-temporal Structure- Since almost every geoscience phenomenon occurs in the realm of space and time, we need to consider evolution of systems in order to understand the current state.

iii. High Dimensionality- The Earth system is incredibly complex, with a huge number of potential variables, which may all impact each other, and thus many of which may have to be considered simultaneously.

iv. Heterogeneity in Space and Time- Geoscience processes are extremely variable in space and time, resulting in heterogeneous datasets in terms of both sparse and clustered data. In addition, the primary evidence for a process may be erased by subsequent processes.

v. Interest in Rare Phenomena- In a number of geoscience problems, we are interested in studying objects, processes, and events that occur infrequently in space and time, such as ore deposit formation and earthquakes.

vi. Multi-resolution Data- Geoscience data sets are often available via different sources and at varying spatial and temporal resolutions.

vii. Noise, Incompleteness, and Uncertainty in Data- Many geoscience data sets are plagued with noise and missing values. In addition, we often have to deal with observational biases during data collection and interpretation.

viii. Small sample size- The number of samples in geoscience data sets is often limited in both space and time, which of course is accentuated by their high dimensionality, (iii) and our interest in rare phenomena (v). In the case examined in this study, the total number of publicly available 3D geological models probably numbers less than 10,000000, and they are stored in a wide variety of formats rendering comparison difficult.

ix. Paucity of Ground Truth- Even though many geoscience applications involve large amounts of data, geoscience problems often lack labelled samples with ground truth.

In this study we specifically focus on six of these challenges by providing a database of one million 3D geological models and resulting gravity and magnetic fields. We address the *Spatio-temporal Structure* of the system by using a kinematic modelling engine that converts a sequence of deformation events into a 3D geological model. We address *High Dimensionality* by generating a very large database of possible outcomes. This represents a fundamental point of difference from many ML targets such as those studying consumer preference or movie rating or facial recognition. Although of course every human face is different, with few exceptions we share the same number of features (eyes, ears, noses), and these features' size and relative positions only varies within small bounds. The number, geometry, composition and relative position of features in the subsurface has very wide bounds and this represents a major hurdle to the application of ML to characterising 3D geology. This challenge is shared by more traditional geophysical inversion approaches (Li and Oldenburg, 1998).

We address issues related to *Multi-resolution Data* by providing a 'controlled' dataset, at the same resolution, it offers possibilities to address multi-resolution issues, by subsampling or upscaling.



We address *Noise, Incompleteness, and Uncertainty in Data* by providing synthetic data, we have noise and uncertainty free
data, or at least under control, and complete spatial coverage over the simulation domain. The models we provide can easily
have a structured or unstructured noise added to them and they can be subsampled to reproduce incomplete datasets.
We address *Small sample size* by generating one million models, which is certainly not enough to thoroughly explore the high-
dimensional model space; however, it illustrates the feasibility of producing large suites of models in the near-future. Modern
ML training sets for popular subjects such as the human face may contain tens of millions of examples (Kollias and Zafeiriou,
2019). A search of the Kaggle database of training datasets (https://kaggle.com, which contains over 63,000 distinct datasets
at the time of writing) only had 151 with geoscience in the keywords, and only seismic catalogues featured as geophysical
data. Similarly, only 59 datasets contained 3D data, and none were related to the geosciences.
Finally, we address the spatial and temporal *Paucity of Ground Truth* by publishing over one million models for which the
full 3D lithological and petrophysical distribution is provided in a labelled form for comparison with resulting gravity and
magnetic fields. This challenge is also faced by geophysical inversion methods. 3D geological models built using sufficient
data to reduce uncertainty arguably exist, but leaving aside a strict definition of uncertainty, well-constrained 3D geological
models are primarily restricted to restricted areas of significant economic interest, specifically sedimentary basins and mineral
deposits, which only represent a sub-set of possible geological scenarios. A number of studies have built simple or complex
synthetic models as a way to overcome these problems by providing fully defined test cases for testing processing, imaging
and inversion algorithms (Versteeg, 1994; Lu et al., 2011; Salem et al., 2014; Shragge et al., 2019a and b). Whilst these
provide valuable insights, the efforts required to build these test cases preclude the construction of large numbers of
significantly different models. It is easy enough to vary petrophysical properties with fixed volumes, however varying the
geometry, and, in particular, the topology is time consuming.
Recent advances in implicit modelling allow extensive geology model suites to be generated by perturbing the data inputs to
the model (Caumon, 2010; Cherpeau et al., 2010; Jessell et al., 2010, Wellmann et al., 2010a & b; Wellmann, and Regenauer-
Lieb, 2012; Lindsay et al., 2012; Lindsay et al., 2013a and b; Lindsay et al., 2014; Wellmann et al., 2014; Wellmann et al.,
2017, Pakyuz-Charrier et al., 2018 a &b, 2019) as part of studies that characterised 3D model uncertainty, however since they
use a single model as the starting point for the stochastic simulations, these works do not provide a broad exploration of the
range of geological geometries and relationships found in nature. Work on the automating of modelling workflows may allow
us to explore the model uncertainty space more efficiently (Jessell et al., 2020).
In this study, we have created a massive open-access resource consisting of one million three-dimensional geological models
using the Noddy modelling package (Jessell, 1981; Jessell & Valenta, 1996). These are provided as the input file that defines
the kinematics, together with the resulting voxel model and gravity and magnetic forward- modelled response. The models
are classified by the sequence of their deformation histories, thus addressing a temporal *Paucity of Ground Truth*. This resource
is provided to anyone who would like to train a ML algorithm to understand 3D geology and the resulting potential field
response, or to anyone wishing to test the robustness of their geophysical inversion techniques. Guo et al. (2021) used the
same modelling engine to produce more than three million models of a more restricted range of parameters to train a ML
Convolutional Neural Network system to estimate 3D geometries from magnetic images. In this study we aim to provide a
much broader range of possible geological scenarios as the starting point for a more general exploration of the geological
model space.
The Noddy software has been used in the past for a range of studies due to its ease in producing 'reasonable-looking'
geological models with a low design or computational cost. A precursor to this study used a hundred or so manually specified
models as a way of training geologists in the interpretation of regional geophysical datasets by providing a range of 3D
geological models and their geophysical responses (Jessell, 2002). Similarly, Clark et al. (2004) developed a suite of ore
deposit models and their potential-field responses. The automation of model generation using Noddy was first explored using



a Genetic Algorithm approach to modifying parameters as a way of inverting for potential-field geophysical data, specifically
gravity and magnetics (Farrell et al., 1996). Wellmann et al. (2016) developed a modern Python interface to Noddy to allow
stochastic variations of the input parameters to be analysed in a Bayesian framework. Finally Thiele et al. (2016 a,b) used this
ability to investigate the sensitivity of variations in spatial and temporal relationships as a function of variations in input
parameters.

In this study we draw upon the ease of generating stochastic model suites to build a publicly accessible database of one million
3D geological models and their gravity and magnetic responses.
**2. Model construction**
The Noddy package (Jessell, 1981; Jessell & Valenta 1996) provides a simple framework for building generic 3D geological
models and calculating the resulting gravity and magnetics responses for a given set of petrophysical properties. The 3D model
is defined by superimposing user-defined kinematic events that represent idealised geological events, namely base stratigraphy
(STRAT), folds (FOLD), faults (FAULT), unconformities (UNC), dykes (DYKE), plugs (PLUG), shear zones (SHEAR-
ZONE) and tilts (TILT), which, can be superimposed in any order, except for STRAT, which can only occur once and has to
be the first event. 3D geological models are calculated by taken the current x,y,z position of a point and unravelling the
kinematics (using idealised displacement equations) until we get back to the time when the infinitesimal volume of rock was
formed, whether defined by the initial stratigraphy, or the time of formation of a stratigraphy above an unconformity, or an
intrusive event. In this study, we only use the resulting voxel representation of the 3D geological models, however it is possible
to produce iso-surface representations of the pre-deformation location of points in an implicit scheme. We have used this tool
as it is rapid, taking under 15s to generate 200x200x200 voxel models with both geological and geophysical representations
combined using an Intel(R) Xeon(R) Gold 6254 CPU @ 3.10GHz processor, and produces 'geologically plausible' models
that may occur in nature. Given that the final 3D model depends on the user's choice of a geological history, Noddy can be
thought of as a kinematic, semantic, implicit modelling scheme.
As opposed to Wellmann et al. ((2016),), Thiele et al. (2016) and Guo et al. (. (2021), who used a python wrapper to generate
stochastic model suites, in this study we have modified the C code itself to simplify use by third parties, although the
philosophy of model generation is an extension of, but very similar to, these earlier studies.
Figure 1 shows one example model set for a STRAT-TILT-DYKE-UNC-FOLD history, consisting of a 3D visualisation
looking from the NE of the voxel model, with some units rendered transparent for clarity, the top surface of the model an EW
section at the northern face of the model looking from the south, a NS section on the western face of the model looking from
the east, and the resulting gravity and magnetic fields.
**3. Choice of Parameters**
In this section we describe the choices and range of values for the parameters that we have allowed to vary for our one million
model suite. We recognise there are other unused modes of deformation that Noddy allows that have been ignored. The
selection of these parameters is based on assessing the range of parameter values that will produce suites of models that we
believe will help and not hinder addressing the challenges cited in the introduction to this work. For example, we limited the
size of the plugs so that a single plug could not replace the geology of the entire volume of interest. In the discussion, we refer
to additional event parameters that could be activated in future studies. We limited the study to five deformation events,
starting with an initial horizontal stratigraphy which is always followed by tilting of the geology. The following three events



are drawn randomly and independently from the event list comprised of folds, faults, unconformities, dykes, plugs, shear
zones and tilts. The likelihood of folds, faults and shear-zones are double the other events as we found that they had a bigger
impact of changing the overall 3D geology, and hence we wished to sample more of these events. This means we can have
$7^3$=343 distinct deformation histories, although the specific parameters for each event can also vary, so the actual
dimensionality of the system is much higher. For clarification, the one million models are not the result of a combinatorial
approach, but of one million independent draws using a Monte Carlo sampling of the model space.
The initial stratigraphy as well as new, above-unconformity stratigraphies, are defined to randomly have between two and five
units to keep the systems relatively simple, but this could of course be increased if desired. The lithology of each unit in a
stratigraphy is chosen to be coherent with the specific event and other units in the same sequence, so that we do not, for
example, mix high-grade metamorphic lithologies and un-metamorphosed mudstones in the same stratigraphic series (Table
2) nor do we assign the petrophysical properties of a sandstone to an intrusive plug. Once a lithology is chosen, the density
and magnetic susceptibility is randomly sampled from a table defining the Gaussian distribution of properties (linear for
density, log-linear for magnetic susceptibility) for that rock type. In the case of densities this may result in occasional negative
values, however since the gravity field is only sensitive to density contrasts this does not invalidate the calculation. Some rock
types have bimodal petrophysical properties to reflect real-world empirical observations, so we draw from a Gaussian mixture
in these cases. The petrophysical data is drawn from aggregated statistics (mean and standard deviation of one or two peaks)
of the approximately 13,500 sample British Columbia petrophysical database (Geoscience BC, 2008).
The parameters which can be varied for each type of event, together with the range of these parameters, is shown in Table 1.
These parameters can be grouped in the shape, position, scale and orientation of the events, and for a five-stage deformation
history require the random selection of a minimum of 23 parameters for a STRAT-TILT- TILT - TILT - TILT model up to 69
parameters for a STRAT-TILT-UNC-UNC-UNC model where each stratigraphy has five units. Apart from the petrophysical
parameters, all other parameters are randomly sampled from a uniform distribution.
Any subset of the geology can be calculated for any sub-volume of an infinite Cartesian space using Noddy, but we limit
ourselves to a 4x4x4 km volume of interest in this study. Similarly, although the geology within this volume can be calculated
at an arbitrary resolution, we have chosen to sample it using equant 20 m voxels as this is well below the typical resolved
measurement scale for these types of data when collected in the field.

Geophysical forward models were calculated using a Fourier Domain formulation using reflective padding to minimise (but
not remove) boundary effects. The forward gravity and magnetic field calculations assume a flat top surface with a 100 m
sensor elevation above this surface, and the Earth's magnetic field with vertical inclination, zero declination and an intensity
of 50,000 nano-tesla.
**4. Results**
The $7^3$ possible event histories produce 343 possible sequences which averages toto 2915 models per sequence. Given the
imposed bias towards folds, faults and shear zones, and the high-probability event sequence (FAULT-SHEAR ZONE-FOLD)
produced 8245 models and the low-probability event sequence (UNC-TILT-PLUG) produced only 905 models, with plateaux
in the number of models calculated giving event sequence frequencies at around 1000, 2000, 4000 and 8000 depending on the
number (0,1,2,3 respectively) of events in the sequence. Together these form a "Noddyverse" of one million 3D geological
models and their gravity and magnetic responses. Figure 2 shows an arbitrarily selected suite of 100 models as a 10x10 grid
showing the top surface and two sections of the model as in Fig 1, together with the resulting gravity and magnetic fields, to
show the variability of the results.

## 5. Applications


The same logic of using millions of Noddy models was first applied by generating a massive 3D model training set and used
to invert real-world magnetic data (Guo et al. 2021). That study used a model suite consisting of only FOLD, FAULT and
TILT events, and only one of each to predict 3D geology using a Convolutional Neural Network. This approach corresponds
to a use case where prior geological knowledge as to the local geological history has been used to limit the model search space,
and formal expert elicitation could provide an important pre-cursor step to support the generation of sensible and tractable
problems (citations). In addition to the CNN training demonstrated by Guo et al. (2021), we can envisage three broad
categories of studies that could build upon the 3D model database we present here:
1) **Studies into the uniqueness of 3D models relative to geological event histories.** The principal question here is
whether any form of clustering of the geophysical fields, and perhaps the map of the surface, can recover the event
sequence or event parameters. Feature extraction techniques are well-known for supporting image classification
and clustering, so using the same principles, can we identify unique clusters of forward models from the
Noddyverse, and do these clusters then correspond to distinct histories? Likewise, can we train a classifier with
extracted features from the forward models of the gravity and magnetic responses which can then successfully
identify models with similar or the same histories. Three broad aspects need to be considered here: (1) the feature
extraction method; (2) choice of pre-processing methods for dimensionality reduction (Self Organising Maps,
Principal Component Analysis, Kernel-Principal Component Analysis, t-distributed Stochastic Neighbor
Embedding etc.) and (3) the clustering (k-means, hierarchical methods, DBSCAN /OPTICS) or classification
methods (random forests, support vector machines, linear classifiers).
A study of geophysical image variability using a simple 2D correlation or maximal information coefficient between
pairs of images of different histories would be illuminating. Do we have images which are the same (or at least very
similar and within the noise tolerance of the geophysical fields) to each other, but belong to very different histories?
If these exist, the ambiguity of the histories can be examined, and we then know where we would expect poor
performance from ML techniques which rely on easily discriminated images. The systems of equations characterising
geophysical inverse problems often? have a non-unique solution. In ML research, if we only use magnetic data or
gravity data for inversion, we will be troubled by the non-uniqueness of the solution. However, because we have both
gravity data and magnetic data, we can extract features from multi-source heterogeneous data at the same time, and
then classify or regress after feature fusion. This could greatly reduce the influence of the non-unique solution.
Having a large set of models will allow clustering of models accordingly to their geophysical response and identifying
subsets of geological models that are geophysically equivalent and cannot be distinguished using geophysical data.
The analysis of diversity of such subsets of models will give an estimate of the severity of non-uniqueness and allow
the derivation of posterior statistical indicators conditioned by geological plausibility.
2) **Comparison between and training of ML systems.** We see potential applications of deep learning techniques
(e.g., Convolutional Neural Network and Generative Adversarial Networks) where the series of models we propose
may also be complemented by other datasets. In this broad topic we would seek to understand which ML
techniques are suitable and effective in mapping geophysical data back to the geology or geological parameters.
We can see potential for investigating which techniques minimise the amount of data necessary to get a good
constraint, i.e., the model structures that most successfully capture geological expert knowledge? This could be
framed as an open challenge to allow different groups to use their preferred approach to the inversion problem.
3) **Validation of the robustness of geophysical inversion schemes**. As previously mentioned, one of the limitations
to validating geophysical inversion schemes is the small number of test models available, with the resulting danger





that the inversion parameters are tuned to the specifics of the test model, rather than being generally applicable. The
Noddyverse model suite allows researchers to trial their inversions against a wide range of scenarios. It will also
allow the examination of the validity and generality of hypotheses at the foundation of several integration and joint
inversion procedures. One well-known example is the underlying assumption that the underlying models vary
spatially in some coherent fashion (Haber and Oldenburg, 1997; Gallardo and Meju, 2003; Giraud et al., 2021).
The analysis of geophysically equivalent models will also enable us to estimate how significantly joint inversion or
interpretation can reduce the non-uniqueness of the solution, with the potential to identify families of geological
scenarios more suited to joint inversion than others. It is obvious that some 3D geological models will be
geologically more complex than others, and that some could be used for the benchmark of deterministic
geophysical inversion of gravity and magnetic data, but also of other geophysical techniques relying on wave
phenomena.

## 6. Discussion

In this study we have produced a ML training dataset that attempts to address four recognised limitations of applying ML to
geoscientific datasets, namely *Spatio-temporal Structure*, *High Dimensionality*, *Small sample size* and *Paucity of Ground
Truth*. Contrary to the current trend, the work for the generation of a comprehensive suite of geological models did not depend
on the appropriate training of a neural network. We relied solely on geoscientific theory and principles while remaining
computationally efficient. While realistic-looking suites of geological models have been generated using Generative
Adversarial Networks (Zhang et al., 2019), these are generally limited to a several thousands of samples, within a limited
range of geological scenarios.

### 6.1 Spatiotemporal Structure

Noddy is by design a Spatio-temporal modelling engine that uses a geological history to generate a model. Simple variations
in the ordering of three events following two fixed events (STRAT & TILT), even with fixed parameters quickly demonstrates
the important of relative time ordering to final model geometry (Fig. 3). While Noddy is limited to simple sequential events,
nature presents geological processes to be coeval (such as syn-depositional faulting) or partially overlapping resulting in
complex spatiotemporal relationships (Thiele et al., 2016a). Nonetheless, re-ordering only sequential events still produces a
vast array of plausible geometries, and indicates the enormity of the model space, and the necessity of efficient methods to
explore them.

### 6.2 High Dimensionality

We have limited ourselves to five deformation events in this study, and no more than five units in any one stratigraphy. These
decisions were based on an idea to "keep it simple" whilst simultaneously allowing a great variety of models to be built. We
recognise that these are somewhat arbitrary choices. We could have true randomly complex 3D histories, leading to models
with, for example, nine phases of folding, however the utility of over-complicating the system is not clear, and would rarely
or ever be discernible in natural systems. Similarly, we limited the parameter ranges of each deformation event, again on the
basis that the ranges chosen made models that are more interesting. For example, there did not seem much interest in having
folds with very large wavelengths or very low amplitudes, as they are equivalent to small translations of the geology and
would translate in the geophysical measurements into a regional trend that is often approximated and removed from the
measurements.





Noddy is capable of predicting continuous variations in petrophysical properties, including variably deformed magnetic
remanence vectors and anisotropy of susceptibility, or densities that vary away from structures to simulate alteration patterns,
however we decided to limit this study to simple litho-controlled petrophysics, whilst recognising the interest of studying
more complex discrete-continuous systems. The indexed models could also be reused with different, simpler petrophysical
variations, such as keeping constant values for each rock type. Each model comes with the history file used to generate the
model and this provides the full label for that model, so that if additional information, such as the number of units in a series
is considered to be important, this can be easily extracted from the file.
**6.3 Small sample size**
The total number of models sounds impressive, however once we divide that number by the 343 different event sequences,
we are left with between 905 and 8245 models per sequence, which whilst still large is by no means exhaustive. There is no
fundamental problem with building 10 or 100 million models, and if this is found to be necessary to provide useful ML training
datasets we can certainly do so at the expense of an increased compute time: these models were built in around a week on a
computer using 20 processor cores. We can also follow try to apply a metric, such as model topology, to analyse how well
sampled the model space is. Thiele et al. (2016b) analysed the topology of stochastically generated Noddy models and found
that after 100 models for small perturbations around a starting model, the number of new topologies dropped off rapidly. In
our case we are not making small perturbations, so we could expect to require more models before the rate of production of
new topologies decays, and topology is only one possible metric for comparing models.
**6.4 Paucity of Ground Truth**
The primary goal of this study was to build a large dataset to provide a wide range of possible models for use in training ML
systems and to test more traditional geophysical inversion systems. The models here, whilst simpler than the large test models
mentioned earlier, represent to our knowledge the largest suite of 3D geological models with resulting potential field data and
tectonic history, which has its own utility. This usage applies equally well to classical geophysical inversion codes, which
have traditionally been tested on only a handful of synthetic models prior to being applied to real-world data, for which there
is no ground truth available.
**6.5 Expert Elicitation**
To use this suite of models as the starting point for inversion of real-world datasets (as has been pioneered by Guo et al., 2021)
we can envisage the introduction of expert elicitation methods to meaningfully constrain the model output space while
acknowledging our inherent uncertainty regarding the model input space. As a probabilistic encoder of expert knowledge,
formal elicitation procedures (O'Hagan, 2006) have contributed greatly to physical domain sciences where complex models
are essential to our understanding of the underlying processes. From climatology/meteorology/oceanography (Kennedy,
2008), to geology and geostatistics (Walker, 2014, and Lark et al., 2015), to hydrodynamics and engineering (Astfalck et al.,
2018, and Astfalck et al., 2019), the central role of expert elicitation is being increasingly recognised. The complexity and
parameterizations of geophysical models, and the expert knowledge that resides within the geophysical community, suggests
this domain should be no different. It is worth noting that the choice of parameter bounds used to define the 1 million model
suite in this article is itself an informal expression of expert elicitation.

**6.6 Extending to the model suite**
In the future we may need a better representation of the "real world" 3D model space, specifically to:



• Include more parameters from Noddy, especially for parameters such as fold profile variation, alteration near structures
to allow petrophysical variation within units. This would help to address the Karpatne et al. (2017) challenge of *Objects*
*with Amorphous Boundaries.* These are capabilities that exist within Noddy but are not used in this study.
• Allow more events to increase the range of outcomes. We arbitrarily restricted ourselves to two started events (STRAT
and TILT) followed by three randomly chosen events, and an extension to the model suite could consider any number
of events in the sequence.
• Include magnetic remanence and anisotropy effects. At present we only model scalar magnetic susceptibility but the
Noddy modelling engine can calculate variable remanence and anisotropic magnetic susceptibility as well.
• Allow linked deformation events. At the moment every event is independently defined, however we could allow
parallel fault sets or dyke swarms, situations which commonly occur in nature.
• Predict different types of geophysical fields. For example, the SimPEG package (Cockett et al., 2015) could easily be
linked to this system to predict electrical fields (Cockett et al. 2015).
• Model larger volumes as large, or deep features cannot currently be modelled due to the 4 km model dimensions.
• Build more models. We in no way believe we have explored the range of possible models in the present model suite,
and if we start in include more events, or more complex event definitions, we will certainly have to generate many more
models, perhaps orders of magnitude more, in order to provide robust training suites and inversion scenarios.
• Add noise to the petrophysical models and/or the resulting geophysical responses. This would help to address the
Karpatne et al. (2017) challenge of *Noise, Incompleteness, and Uncertainty in Data. Incompleteness* can be addressed
by removing parts of the geophysical data and does not require new models to be built. Similarly, the challenge of
*Multi-resolution Data- Geoscience* could be addressed by subsampling parts or all of existing geophysical outputs.
• Include topographic effects. In this study, we have ignored the effect of topography on the models, although again this
could be included in the future, as it is supported by Noddy.
We also need to be clear that a model built in Noddy is not capable of predicting all geological settings, as all Noddy models
are plausible geology, but not all plausible geology can be modelled by Noddy. To improve this situation, we would need to
improve the modelling engine itself. Similarly, the logic of trying to predict geology from geophysical datasets in this study
is only partially fulfilled: the geometry comes from geological events sequence, but identical geometries can be produced by
different event sequences.
**7. Conclusions**
This study represents our first steps in producing geologically reasonable training sets for ML and geophysical inversion
applications. We have used Noddy to generate a very large, open-access 1M model, set of 3D geology and resulting gravity
and magnetic models as a ML training sets. These training sets can also be used as test cases for gravity and/or magnetic
inversions. The work presented here may be a first step to overcoming some of the fundamental limitations of applying these
techniques to natural geoscientific datasets.



## 8. Acknowledgements

We acknowledge the support from the ARC-funded Loop: Enabling Stochastic 3D Geological Modelling consortia (LP170100985), DECRA (DE190100431) and Data Analytics for Resources and Environment ITTC (IC190100031). The work has been supported by the Mineral Exploration Cooperative Research Centre whose activities are funded by the Australian Government's Cooperative Research Centre Programme. This is MinEx CRC Document 2021/***. This work was further supported by the ARC Data Analytics for Resources and Environments ITTC (IC190100031). We would like to thank AARNET for supporting this work by hosting the 500GB model suite at CloudStor.

## 9. Code and Data availability

A doi (https://zenodo.org/record/4589883) provides access GitHub repository which contains the following elements (Jessell, 2021):

1. The source code (C language) for the version of noddy adapted to producing random models.

2. A readme.md file with a link to the windows version of the Noddy software, plus a link to 343 tar files, one for each event history ordering of the model suite.

3. A Jupyter Notebook (python code) for sampling from and unpacking the models.

4. A link in the same readme.md file to the equivalent *mybinder.org* version of the notebook so that no code installation is required to sample from and view the model suite: https://mybinder.org/v2/gh/Loop3D/noddyverse/HEAD?filepath=noddyverse-remote-files-1M.ipynb

All codes and data are released under the MIT licence.

## 10. Author Contribution

Mark Jessell wrote the original and modified noddy software, ran the experiments and wrote the python software for visualising the models. Jiateng Guo and Yunqiang Li were in volved in conceptualisation and manuscript preparation. Mark Lindsay, Jérémie Giraud and Guillaume Pirot were involved in the conceptualisation, as well as in co-writing the introduction and discussions sections of the paper. Vitaliy Ogarko, Richard Scalzo and Ed Cripps were involved in developing and co-writing the introductions and discussion sections of the manuscript.

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

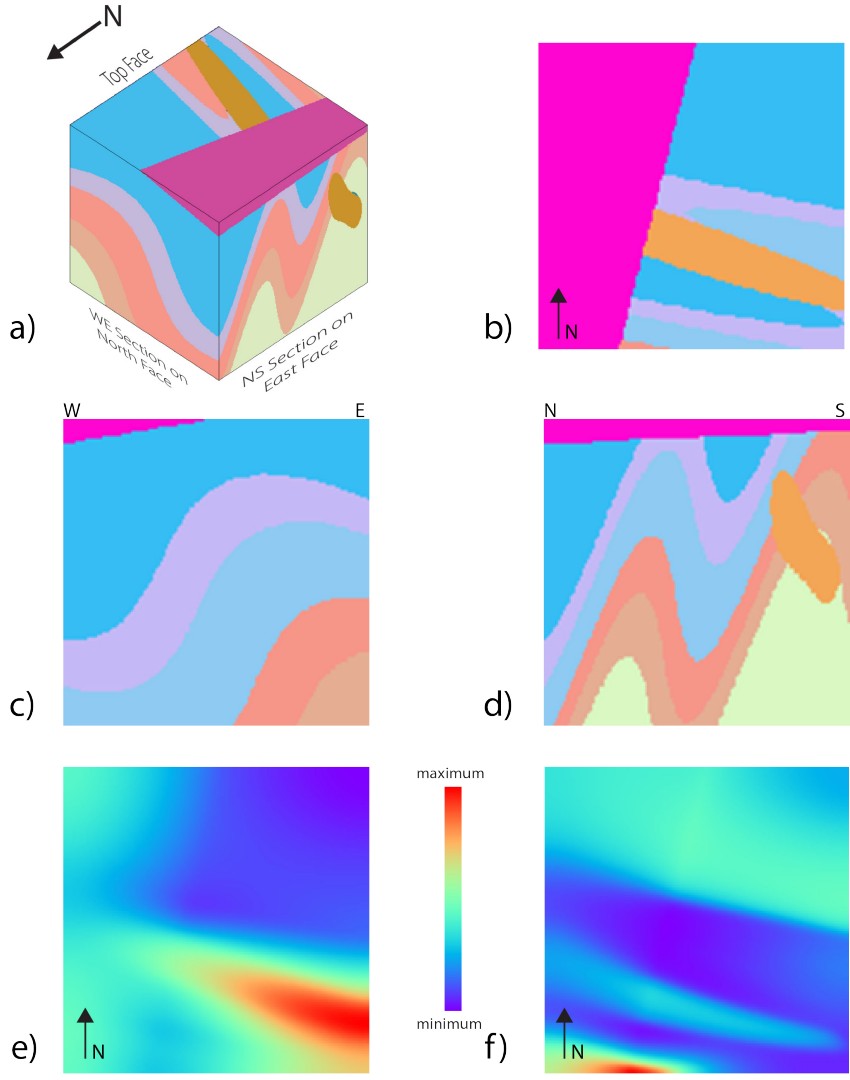


**Figure 1. Example model set for a STRAT-TILT-DYKE-UNC-FOLD sequence showing a) 3D visualisation looking from the NE of the voxel model, b) the top surface of the model, c) an EW section at the northern face of the model looking from the south, d) a NS section on the western face of the model looking from the west, and the resulting e) gravity and f) magnetic fields. Geophysical images are all normalized to model max-min values.**

475

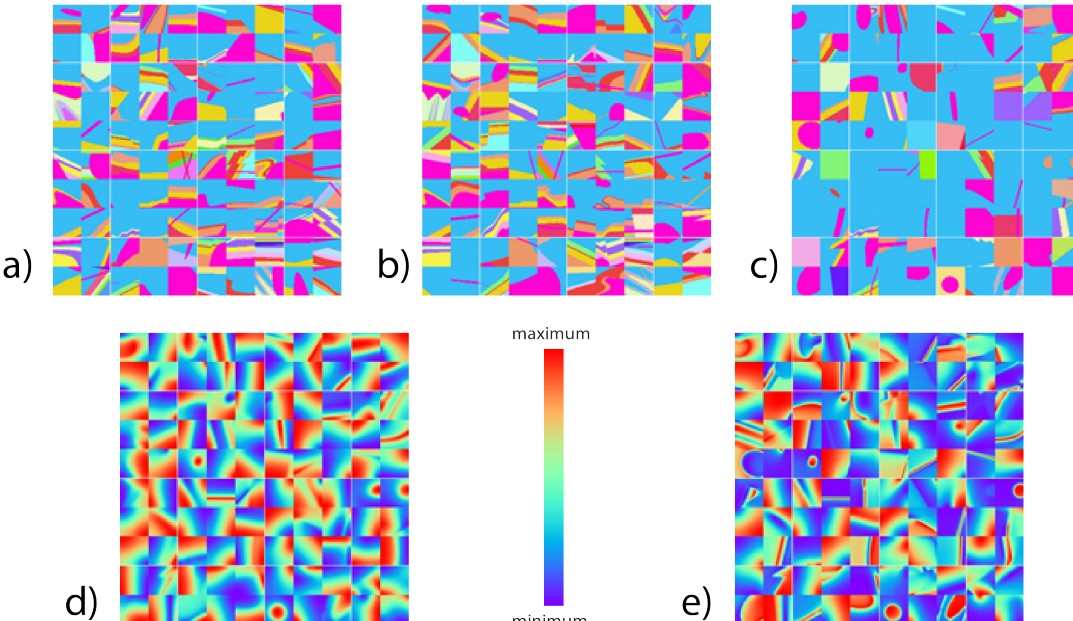

**Figure 2. Example models for 100 randomly selected models drawn from the 1M model suite showing a) the top surface of the model, b) an EW section at the northern face of the model looking from the south, c) a NS section on the western face of the model looking from the west, and the resulting d) gravity and e) magnetic fields. Geophysical images are all normalized to model max-min values.**


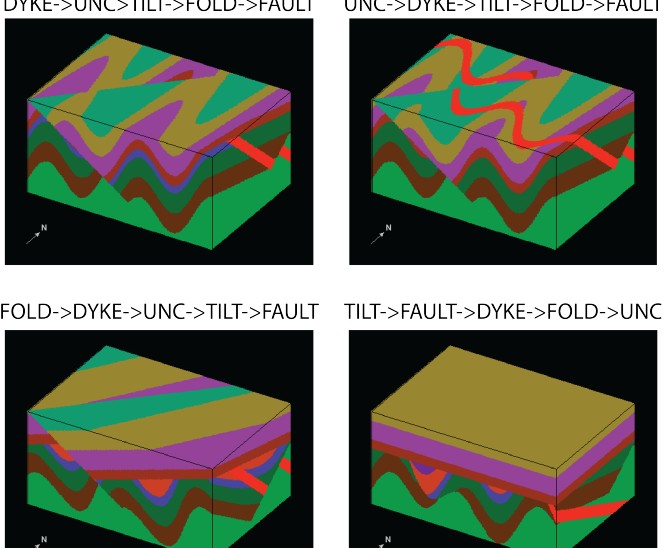


**Figure 3. Four possible 3D geological models with the same base stratigraphy (STRAT) followed by five events using four of the**
**possible different event ordering sequences.**





| Event type | Parameter 1 | Parameter 2 | Parameter 3 | Parameter 4 | Parameter 5 | Parameter 6 | Min/Max number of parameters |
|---|---|---|---|---|---|---|---|
| Base Stratigraphy | Number of units. Range: 2-5 | unit n thickness: 50-1000 m | Density of each unit: depends on lithology of unit n | Magnetic susceptibility of each unit: depends on lithology of unit n | | | 5/12 |
| Fold | Wavelength: 1,000-11,000 m | Amplitude: 200 - 5,000 m | Azimuth: 0-360 degrees | Inclination: 0-90 degrees | Phase: 0-4000 m | Along axis amplitude decay: 500-9,500 m | 6/6 |
| Fault | Position of 1 point on fault: x,y,z 2000-4000 m | Displacement: 0-2000 m | Azimuth: 0-360 degrees | Inclination: 0-90 degrees | Pitch of displacement: 0-90 degrees | | 7/7 |
| Unconformity | Position of 1 point on Unconformity: x=2000-3000m y=2000-4000m z=3000-4000 m | Number of units above unconformity: 2-5 | Azimuth: 0-360 degrees | Inclination: 0-90 degrees | Density of each unit: depends on lithology of unit n | Magnetic susceptibility of each unit: depends on lithology of unit n | 10/17 |
| Dyke | Position of 1 point on fault: x=0-4000 m y=0-4000 m z=0-4000 m | Azimuth: 0-360 degrees | Inclination: 0-90 degrees | Width of Dyke: 100-400 m | Density: depends on lithology | Magnetic susceptibility: depends on lithology | 8/8 |
| Plug | Shape: Cylindrical, Conic, Parabolic, Ellipsoidal | Position of centre of plug: x=1000-4000m y=1000-4000m z=1000-4000m | Size of plug: parameter varies with shape | Density: depends on lithology | Magnetic susceptibility: depends on lithology | | 7/9 |
| Tilt | Position of 1 point on rotation axis: x=2000-3000m y=2000-4000m z=3000-4000m | Azimuth: 0-360 degrees | Inclination: 0-90 degrees | Rotation: -90-90 degrees | | | 6/6 |
| Shear zone | Position of 1 point on fault: x,y,z 2000-4000 m | Displacement: 0-2000 m | Azimuth: 0-360 degrees | Inclination: 0-90 degrees | Pitch of displacement: 0-90 degrees | Width of Shear Zone: 100-2000 m | 8/8 |

Table 1. Free parameters with their allowable ranges for each event.






| Lithology | Lithology Class | Genetic Class | Mean Density g.cm-3 | Standard Deviation Density | Mean Log Susceptibility (cgs) | Standard Deviation Log Susceptibility | Susceptibility Bimodality Flag |
|---|---|---|---|---|---|---|---|
| Felsic_Dyke_Sill | Dyke | Intrusive | 2.612593 | 0.090526329 | -3.693262 | 1.50094258 | 1 |
| Mafic_Dyke_Sill | Dyke | Intrusive | 2.793914 | 0.015759637 | -2.119223 | 0.85376583 | 0 |
| Granite | Plug | Intrusive | 2.691577 | 0.094589692 | -2.455842 | 0.86575449 | 1 |
| Peridotite | Plug | Intrusive | 2.851076 | 0.154478049 | -1.158807 | 0.4390425 | 0 |
| Porphyry | Plug | Intrusive | 2.840024 | 0.128971814 | -2.613833 | 0.99194475 | 1 |
| Pyxenite_Hbndite | Plug | Intrusive | 3.194379 | 0.253322535 | -1.946615 | 1.03641373 | 0 |
| Gabbro | Plug | Intrusive | 3.004335 | 0.159718751 | -2.124022 | 0.82126305 | 1 |
| Diorite | Plug | Intrusive | 2.851608 | 0.134656746 | -2.088111 | 0.81829275 | 1 |
| Syenite | Plug | Intrusive | 2.685824 | 0.115078068 | -2.461453 | 0.91295395 | 1 |
| Amphibolite | Met_strat | Metamorphic | 2.875933 | 0.142164171 | -2.69082 | 0.90733619 | 1 |
| Gneiss | Met_strat | Metamorphic | 2.701191 | 0.073583537 | -3.18094 | 0.95259725 | 1 |
| Marble | Met_strat | Metamorphic | 2.871775 | 0.532997473 | -3.671996 | 1.25374051 | 0 |
| Meta_Carbonate | Met_strat | Metamorphic | 2.738965 | 0.036720136 | -3.117868 | 0.82945531 | 0 |
| Meta_Felsic | Met_strat | Metamorphic | 2.782584 | 0.301451931 | -3.55755 | 0.65748564 | 1 |
| Meta_Intermediate | Met_strat | Metamorphic | 2.894892 | 0.265153614 | -3.673276 | 0.26107008 | 0 |
| Meta_Mafic | Met_strat | Metamorphic | 2.814461 | 0.096381942 | -3.250044 | 0.62513286 | 0 |
| Meta_Sediment | Met_strat | Metamorphic | 2.982992 | 0.49439556 | -3.402807 | 0.89505466 | 1 |
| Meta_Ultramafic | Met_strat | Metamorphic | 2.843941 | 0.138208079 | -2.166206 | 0.76543947 | 0 |
| Schist | Met_strat | Metamorphic | 2.81978 | 0.109752597 | -3.18525 | 0.69584686 | 0 |
| Andesite | Met_strat | Volcanic | 2.721189 | 0.091639014 | -2.15826 | 0.71678329 | 0 |
| Basalt | Met_strat | Volcanic | 2.79269 | 0.155153198 | -2.155728 | 0.64718503 | 0 |
| Dacite | Met_strat | Volcanic | 2.62127 | 0.129131224 | -2.562422 | 0.8166926 | 0 |
| Ign_V_Breccia | Met_strat | Volcanic | 2.910459 | 0.101746428 | -2.706956 | 0.73116944 | 0 |
| Rhyolite | Met_strat | Volcanic | 2.630833 | 0.071233818 | -3.046728 | 0.78711701 | 0 |
| Tuff_Lapillistone | Met_strat | Volcanic | 2.64447 | 0.110173772 | -2.878701 | 0.86889142 | 0 |
| V_Breccia | Met_strat | Volcanic | 2.771579 | 0.167796457 | -2.524945 | 0.90943985 | 0 |
| V_Conglomerate | Met_strat | Volcanic | 2.755267 | 0.10388303 | -2.304483 | 1.00991116 | 0 |
| V_Sandstone | Met_strat | Volcanic | 2.779715 | 0.101133121 | -2.903361 | 0.82701019 | 0 |
| V_Siltstone | Met_strat | Volcanic | 2.859347 | 0.102741619 | -2.769054 | 0.87771183 | 0 |
| Conglomerate | Strat | Sedimentary | 2.618695 | 0.116158268 | -3.31026 | 0.9740717 | 0 |
| Limestone | Strat | Sedimentary | 2.713912 | 0.147683486 | -4.256256 | 0.87772406 | 0 |
| Pelite | Strat | Sedimentary | 2.698554 | 0.021464631 | -3.369295 | 0.5295974 | 1 |
| Phyllite | Strat | Sedimentary | 2.739177 | 0.173374383 | -3.696455 | 0.73955588 | 0 |
| Sandstone | Strat | Sedimentary | 2.622672 | 0.107003083 | -3.452758 | 0.64521521 | 0 |
| Greywacke | Strat | Sedimentary | 2.861463 | 0.16024622 | -3.841047 | 1.14724626 | 1 |


**Table 2. Simplified petrophysical values derived from British Columbia database (Geoscience BC, 2008). Values are randomly**
**sampled from Gaussian distributions defined by mean and standard deviation of density and log magnetic susceptibility. For**
**lithologies with bimodal magnetic susceptibilities (flag=1), mixed sampling is based on offsetting the means by +/-0.75 orders of**
**magnitude, which approximates the variations seen in nature.**