# Peer review of "Into the Noddyverse: A massive data store of 3D geological models for"

_Earth System Science Data, 2021_

## Author Comment (AC1)

*Line numbers in blue refer to the edited version*

- **RC1**: 'Comment on essd-2021-304', Anonymous Referee #1, 26 Oct 2021  reply

**General Comment:**

The manuscript presents a large dataset that focuses on the geometry of the area of interest and can be useful for training machine learning models in geosciences. The dataset is a step forward in providing the data science community with geoscience related dataset. The authors also recommend several use-cases of the dataset and future works. I have a few minor suggestions/comments to improve the quality of the manuscript. I also feel that the quality of writing can be improved. The models/model history are available on Github as described with a convenient Jupyter Notebook for users.

**Detailed Replies:**

In generating a geologic dataset, users typically have specific parameters of interest. Is there a way for users to conveniently select uncertain parameters to test within the generated dataset?

*The Jupyter notebook allows models to be filtered by event sequence, and with modification could be filtered for any parameter. This point is now made in section 6.5.*

*More generally, outside the scope of this manuscript, the pynoddy code provides this option for structural parameters as described in Wellmann et a., 2016, although it does not manage petrophysical variations. We chose to modify the base c language version of Noddy to speed up the calculations. In the future the two codes will probably be merged.*

Line 100: Perhaps the authors should highlight why the publication of a 1-million dataset is needed when users of the Noddy platform can generate up to 3-million models as mentioned by the author in line 100 - in most cases, users typically want to test specific features as opposed to all general features and may have to regenerate the dataset.

*We of course agree that focusing on a subset of structures when the targeted structure is reasonably well characterised is a valid approach, as was taken by Guo et al. 2021, and discussed in section 6.5. Unfortunately, in many parts of the world there is no outcrop available, due to tens to hundreds of metres of cover. In this scenario, it makes sense to start with a broader search for possible 3D models that may match the observed gravity or magnetic response, given their inherent ambiguity. We can imagine a hierarchical approach where a subset of the 1M models is identified as possible causative structures, and then these are accepted or rejected based on the*

*geologist's prior knowledge, and the accepted models are then used as the basis for a focussed parameter exploration. This point is now made in section 6.5.*

Line 148-149: "The likelihood of folds, faults and shear-zones are double the other events as we found that they had a bigger impact of changing the overall 3D geology" - is there a way to illustrate or quantify this?

*This is certainly an interesting question related to the impact of different event types on 3D geology, but for this study it was a qualitative observation, and we believe a quantitative investigation is beyond the scope of this paper. This has been clarified in line 155.*

Line 151-152: Perhaps highlight how the sampling method (combinatorial versus MC) affects the generated models?

*Whilst a combinatorial approach may in theory explore the parameter space more uniformly, the sequence of 5 deformation events is so non-linear that it was reasoned that a pure MC approach would serve our purposes. This point is now made on line 158.*

Line 240: Line 56 says that the focus is on six challenges but here it mentions that the authors attempt to address four recognised limitations.

*The two missing challenges (Multi-resolution Data & Noise, Incompleteness, and Uncertainty in Data) have been added to line 251.*

Line 242-243: Not clear what is meant by "Contrary to the current trend, the work for the generation of a comprehensive suite of geological models did not depend on the appropriate training of a neural network".

*This has been rephrased to make it clear that we do not rely on the manual labelling of datasets on line 252.*

Line 244-246: Worth mentioning that the problem with GAN is not the amount of samples that can be generated (as the sampling process is fast), the quality of generated samples are limited by the number of training samples used, as well as the stability of GAN in generating realistic samples.

*We agree and this point has been added to line 256.*

Line 485: Figure 3 is not called anywhere in the manuscript

*Fig. 3 is in fact called on line 260.*

**Minor Comments:**

*All minor corrections have been applied*

Line 30: "applied" -> "application"?

Line 46: best to be consistent with either "data set" or "dataset"

Line 50: be consistent with capitalization

Line 62: "varies" -> "vary"?

Line 132: Extra parenthesis, "python" -> "Python"?

Line 179: "toto" -> "to"

Line 193: "citations" needs to be updated

Line 211: "often?" needs to be updated

Line 306: "started"?

Line 317: "start in"?

Line 358: "in volved"

---

## Author Comment (AC2)

*Line numbers in blue refer to the edited version*

- **RC2**: 'Comment on essd-2021-304', Jiajia Sun, 01 Nov 2021   reply

The authors created a data set consisting of 1 million geological models and the associated gravity and magnetic responses using the Noddy package. This is very timely and welcome contribution to the geophysical community. It is broad applications for training machine learning models for predicting geology (including history, geometry, structure, etc) and for testing inversion algorithms (e.g., understanding the non-uniqueness of inversion). As the authors mentioned, the geoscience community suffers from a lack of large, labelled datasets that can be used to validate or train robust Machine Learning and inversion schemes. This contribution is a timely and valid response to this problem. As such, it is my belief that the authors' work fills an urgent need in the geoscience community. I would like to commend the authors for recognizing such a critical need and for developing a first-step solution to it.

The authors also discussed three possible applications of this massive data set in Section 5. They are all highly relevant and deserve future research work. This again highlights the importance of the authors' work documented in this manuscript.

I am also glad to see that the authors recognized the limitations in their current work and discussed several ways to expand and improve the repository of 'real world' geological models.

I also tested the notebook (on mybinder.org) and visited the repository https://cloudstor.aarnet.edu.au/plus/s/8ZT6tjOvoLWmLPx. They both work and are in good shape.

I do not have any major concerns. Below are some minor grammatical and/or clarification suggestions and questions.

Detailed comments

Line 52: 'In the case examined in this study, the total number of publicly available 3D geological models probably numbers less than 10,000000,'  What case? Where does this number come from?

*This statement was based on an estimate and does not add much to the discussion so has been deleted.*

Line 59: 'a very large database of possible outcomes'. Not exactly sure how to understand 'outcomes'. Based on the context, I supposed it means geological outcomes of a series of geological events such as faulting, folding, intrusion, etc. Is that correct?

*This has been clarified in the text.*

Line 86: Exactly!

Line 87: How is 'implicit modeling' defined? And how is it different from 'explicit modeling' (if the latter exists)?

*A definition of implicit modelling has been supplied and compared with CAD-style explicit modelling in line 90.*

Line 98-99: Great contribution!

Line 123: 'taken' à 'taking'

*fixed*

Figure 1: Please double check the 3D visualization in panel (a). Looking from NE to SW, the East face should be in the left and the North Face in the right.

*Fixed, it is looking from the NW.*

Line 152: 'Monte Carlo sampling'. From the text below, it seems that only Gaussian and uniform sampling were employed when generating the petrophysical and all the other parameters. Does 'Monte Carlo sampling' simply mean random sampling from Gaussian and uniform distributions?

*In this case yes, and this has been clarified.*

Line 154-157: Great! It is important to make the lithologies consistent with the associated geological events. This is where expert knowledge from geologists can play an irreplaceable role. Just curious about how this was realized. Did the authors develop an automated way of ensuring geological consistency? Manually checking each geological model and evaluating its geological and lithological consistency do not seem practical.

*An explanation of the hierarchical grouping of 'associated lithologies' is provided in line 161.*

Line 181-183: Please rephrase this sentence, as it is very long and hard to follow.

*Sentence has been rephrased*

Line 193: 'citations'? *fixed*

Line 196: 'clustering of geophysical fields'. Did the authors mean classification of gravity and magnetic measurements into different classes?

*Yes, and this has been specified more clearly.*

Line 200: "forward models of the gravity and magnetic response". Not exactly sure what is meant here. Seems to me that 'forward models' is simply a repeat of the 'gravity and magnetic response'. Please rephrase.

*This has been rephrased.*

Line 211: remove the question mark.

*done*

Line 219: Remove the word 'and' in the heading.

*done*

Line 229: suggest replacing 'trial' with 'test'.

*done*

Line 233-235: Excellent!